# Through a Horse’s Eyes: Investigating Cognitive Bias and Responses to Humans in Equine-Assisted Interventions

**DOI:** 10.3390/ani15040607

**Published:** 2025-02-19

**Authors:** Céline Rochais, Emilie Akoka, Suzanne Amiot Girard, Marine Grandgeorge, Séverine Henry

**Affiliations:** 1EthoS (Éthologie Animale et Humaine)—UMR 6552, Centre National de la Recherche Scientifique (CNRS), University Rennes, Normandie University, F-35000 Rennes, France; akomilie@gmail.com (E.A.); s_amiot@yahoo.com (S.A.G.); marine.grandgeorge@univ-rennes.fr (M.G.); severine.henry@univ-rennes.fr (S.H.); 2School of Animal, Plant & Environmental Sciences, University of the Witwatersrand, Johannesburg 2000, South Africa

**Keywords:** animal-assisted interventions, cognitive bias, emotional state, horses, welfare

## Abstract

This study investigated how horses involved in different types of work—specifically, conventional riding school lessons and equine-assisted interventions—perceive environmental stimuli and interact with humans. We investigated the cognitive judgement biases (such as a tendency to be more or less optimistic in ambiguous situations) and the perception of humans (such as a tendency to be more or less aggressive toward humans) in horses from three different facilities. Some horses participated only in riding school lessons (RS), while others were used for both riding school activities and equine-assisted interventions (EAI-RS). We hypothesised that EAI-RS horses would be more negatively impacted than RS horses because two types of work may be demanding. While we found no clear link between the type of work and a more negative view of humans or increased pessimism, EAI-RS horses in facilities with restricted access to food and high workloads appeared more pessimistic and indifferent toward unfamiliar humans. We thus suggest that the additional workload associated with EAIs may have been mitigated by other environmental conditions related to the overall living conditions and facility management.

## 1. Introduction

Perception is a cognitive function that contributes to the construction of mental representations of environmental stimuli. Most behaviours, which are at least partially triggered by external stimuli, rely on perception [1]. How animals perceive their world is a crucial and ethically relevant question because their perception of environmental stimuli can be either more or less positive or negative. Perception can depend on the valence of an individual’s underlying mood, i.e., long-lasting affective state arising from the accumulation and integration of short-lasting emotions in response to specific stimuli [2,3]. Positive mood depends on internal physiological balance, a satisfaction of behavioural needs, an absence of negative emotions, and the presence of positive emotions [4]. In humans, negative mood is associated with cognitive biases such as negative judgement bias (a greater expectation of negative outcomes under ambiguity, i.e., ‘pessimism’) and attention and memory negative bias [3].

The underlying negative mood that induces cognitive pessimistic judgement bias also occurs in non-human animals, such as various mammal, bird, and insect species (reviewed in [5]). Cognitive judgement bias measurement can thus be used to infer the mood of animals using non-verbal cues to better understand their welfare state, defined as “a chronic positive mental and physical state resulting from the satisfaction of the animal’s behavioural and physiological needs and expectations” [6]. Harding et al. [7] was the first to investigate cognitive judgement bias and showed that inducing a negative mood state using a chronic mild stress procedure led to ‘pessimistic’ bias in rats responding to ambiguous auditory stimuli. Harding’s method has now been used in a large number of studies, mostly by comparing stressed and unstressed animals, such as in species-specific standard versus barren conditions [5]. There has been support for a negative mood induction associated with pessimistic judgement bias, but some studies showed no or contrasting results. Beyond the influence of environmental conditions, to our knowledge, there is no study focusing on the influence of an animal’s work/activity on pessimistic judgement bias.

Like humans, some domestic animals are involved in working activities, and hence their mood and cognitive processes may be impacted in their daily life by their working conditions. For example, dogs involved in animal-assisted interventions have to calmly tolerate sometimes invasive interactions and may experience repeated stress that leads to them enduring negative affective states related to apathy, associated with a decrease in overall attention [8]. Several studies have described apathetic states (directed to the general environment, including of humans) in working horses [9] and subsequent changes in cognitive processes (e.g., attention) [10]. There are several explanations for these results, such as facility management, restrictive and painful working conditions, and negative human–animal interactions in horses [11,12,13,14], highlighting that working animals can be under emotional and physical strain.

One particular work/activity comprises animal-assisted interventions (AAIs), defined by the International Association of Human–Animal Interaction Organisations [15] as ‘goal-oriented and structured interventions that intentionally include or incorporate animals in health, education and human services (e.g., social work) for the purpose of therapeutic gains in humans’. AAIs have become increasingly popular [16]. AAIs use various animals, among which horses are one of the most frequently used species, e.g., for interaction with children with autism spectrum disorders [17]. The effects of these equine-assisted interventions (EAIs) on humans have been widely studied, but studies focusing on the animals involved remain scarce (reviewed in [18]). Further studies are thus needed, especially because EAI horses often have to deal with the same constraints in their daily life as other working horses and also with an additional need to cope with EAI activities such as unpredictable behaviours, attention problems, or aggressivity from human participants. Horses also have to deal with persons with posture or balance problem (e.g., asymmetrical weight distribution that can cause lameness or back pain in horses) [19].

Few studies have measured the acute reactions of horses to EAI sessions, and most of them have provided contrasting results or reported that the type of work per se or type of person involved has no particular impact on the horses [20,21,22,23]. However, recent results have shown more “stress” behaviours and higher serum cortisol concentrations in horses ridden in EAI sessions involving participants with autism spectrum disorder, intellectual disabilities, and sensory processing disorders compared with horses ridden by participants with attention deficit hyperactivity disorder, epilepsy, traumatic brain injury, Down syndrome, and cerebral palsy [24]. Conversely, only three studies explored the chronic effects of EAIs on the behaviour of horses outside work sessions, all concerning the reactions of EAI horses in a human–horse relationship test. The reactions of horses in these tests primarily reflect their perception, positive or negative, of humans (for a review [25]). Previous studies have shown that EAI horses were less interactive than riding school or sport horses [26,27]. This outcome highlights a selection for a less reactive temperament in EAI horses, a specific training, or a compromised welfare leading to an apathetic state or lower motivation to interact with humans [26]. Furthermore, a recent study also showed that horses working in both EAIs and RSs appeared more affected than other RS horses, as reflected by their working modalities and greater negativity towards humans [14]. This suggests that EAI activities have chronic effects that may add to other management concerns, but it also shows that there is very little information about the chronic welfare of EAI horses.

In horses, judgement bias tests have demonstrated, for example, that riding school horses subjected to constrained working conditions and living in restricted conditions (spatial, social, feeding restrictions) have pessimistic bias and behavioural indicators of poor welfare, while leisure horses living under more naturalistic life conditions have optimistic bias and behavioural indicators of good welfare [28]. Similarly, horses kept in isolated stalls have positive judgement bias when released in the pasture for a few days with other horses [29], and management comparisons revealed optimistic judgement bias in horses housed in groups in paddocks with free access to roughage [30]. Another study tested the effect of training using positive or negative reinforcement in horses and showed that negative reinforcement training was associated with optimistic judgement bias [31]. Overall, ‘naturalistic’ housing conditions (i.e., conditions aligned with the species-specific needs of horses to live in stable groups with free access to roughage and free movement [32]) are associated with optimistic judgement bias and behavioural indicators of good welfare. In addition to their living conditions, the impact of the type of work on cognitive judgement and perception of humans by horses still needs further investigation.

The aim of the present study was to test horses’ perception of environmental stimuli, including humans, according to the work they perform. We investigated cognitive judgement bias and the perception of humans by horses involved in either only ‘conventional’ riding school (RS) lessons or in both riding lessons and EAI (EAI-RS), to assess how work impacts the chronic mood/welfare of horses. We hypothesised that EAI-RS horses would show a pessimistic bias (expectation of negative outcomes under ambiguity) because they must cope with the same constraints in their daily life as other working horses and also engage with a variety of people with unusual behaviours or postures. We also expected that EAI-RS horses would be less interactive than RS horses during human–horse relationship tests, as a reflection of compromised welfare. Finally, we also predicted that the facility in which the horses were housed may have induced variations in their perception of their environment, including of humans, because facility management is related to the welfare of horses.

## 2. Materials and Methods

### 2.1. Ethical Statement

The experiments were carried out in accordance with the Directive 2010/63/UE of the European Parliament and the Council on the protection of animals used for scientific purposes. They complied with the current French laws related to animal experimentation (decree no 2013 ± 118 of 1 February 2013) and its five implementation orders (JO 7 February 2013, integrated into the Rural Code and the Code of maritime fishing under no R. 214 ± 87 to no R. 214 ± 137). The experiments performed in this study were not within the scope of application of the European directive and thus do not require ministerial authorization (reference #2023022813323100). These experiments involved only behavioural observations and non-invasive interactions with the horses. The horses used in this research were not research animals. Animal husbandry and care were under the management of the riding school staff. The riding school managers provided the authors with their informed consent for this study.

### 2.2. Animals and Study Sites

This study involved three different riding facilities (referred as sites throughout) located in western France. Observations were performed between May 2023 and April 2024 (site 1: May–June 2023; site 2: September–October 2023; site 3: March–April 2024). Site selection was based on the following: (1) the horses’ “naturalistic” housing conditions (i.e., conditions aligned with species-specific needs of horses to live in stable groups with free access to roughage or grass [32]), and (2) horses involved in riding school and EAI activities. All horses involved in riding school and EAI activities (EAI-RS) across the three sites were included in the study. To create a comparable control group, we then selected horses from the same sites exclusively involved in RS activities, ensuring that their individual characteristics (e.g., age, sex) were as close as possible to those of the EAI-RS group.

We studied 30 horses (10 females; 20 geldings; median: 15—Q1–Q3: 13–19 y.o., range: 9–28 y.o.). They were mostly ponies (80% > 148 cm at withers, International Federation for Equestrian Sport) from various breeds, mostly unregistered (Table 1). At the time of the study, they had been in their facility and had been involved in the same working practises for at least one year.

At site 1, horses were kept in pasture all year around and in a stable herd (N = 16 individuals in the herd, 13 of 16 were studied) consisting of the same individuals housed together for more than six months prior to our study. Grass and hay were available ad libitum and horses did not receive any supplementary feed. They had access to a shelter ad libitum.

At sites 2 and 3, housing was more restricted. Pasture time depended on weather conditions and riding lessons. Hay availability was restricted (i.e., facilities adjusted the amount of hay for each horse based on its body weight).

At site 2, horses were kept in a stable herd (N = 20 individuals in the herd, 10 of 20 were studied) consisting of the same individuals housed together since birth. They were housed either in a pasture with grass ad libitum (60% of time), either in a paddock with restricted hay availability provided once per day (30% of time) or in group stalls with straw bedding and hay provided once per day (10% of time). Horses received supplementary homemade pellets once per day (8:30 a.m.).

At site 3, horses were kept in a stable herd (N = 30 individuals in the herd, 7 of 30 were studied) consisting of the same individuals housed together for more than six months prior to our study. They were housed either in a paddock with hay available ad libitum (43% of time), or in pair stalls with straw bedding and hay provided twice per day (57% of time). Horses received supplementary pellets three time per day (8:30 a.m., 12:30 a.m., 6:30 p.m.). At all 3 sites, water was freely accessible through automatic drinkers.

### 2.3. Working Conditions

Overall, 14 horses were involved in conventional riding school lessons (RS, 6 mares, 8 geldings, median = 14 y.o.) and 16 horses were involved in both RS and EAI activities (EAI-RS, 4 mares, 12 geldings, median = 19 y.o., Table 1). At each site, horses were kept in the same overall living conditions regardless of their type of work. RS and EAI sessions took place in an indoor arena at all sites. Horses worked between 1.5 and 11 h a week. The proportion of EAI work for EAI-RS horses ranged from 9% to 86% of their working time (mean ± SEM = 31.4 ± 6.9% of the total amount of working hours per week).

EAI work consisted of grooming, saddling, mounting, riding, and dismounting, and varied between riding sites. At each site, riding consisted of learning basic riding elements while walking and performing exercises (rotating/bending, outstretching upper arms) with games such as rods, cones, or balls. Riders were children and teenagers with multiple disabilities, ASD (mostly in riding sites 2 and 3), or social problems. EAI sessions involved the riding teacher handling a horse, with a horse handler and several therapists in charge of the people and handling horses if necessary. Site 1 was characterised by a low amount of work per week and a large EAI load (from 28% to 86% of the horses’ working time). Horses were involved in EAI sessions 1 to 5 times a week. EAI sessions mostly lasted 1 h and consisted of 5 min mounting (with or without a lift machine, according to the disability of the rider), 45 min riding, 5 min dismounting, and 5 min grooming. Horses were groomed and saddled by handlers 5 min prior to rider engagement and there was no warming up. Sites 2 and 3 were characterised by a higher amount of work compared to site 1 (Table 1) and a low EAI load. Horses were involved in one EAI session per week. Thus, EAI work comprised from 9% to 17% of the horses’ working time. EAI sessions lasted 2 h and consisted of 15 min grooming, 10 min saddling, 5 min mounting (without a lift), 1 h riding, 5 min dismounting, 10 min unsaddling, and 15 min grooming.

RS work followed the same protocols as EAI (from grooming to grooming). Riders were beginners—children and teenagers. RS sessions involved the same riding teacher but no handler. At site 1, RS sessions occurred 1 to 7 times a week, lasted 1 h, and consisted of leisure riding and exercises (e.g., slalom). In contrast to EAI, RS riders groomed and saddled horses prior to and after riding. At sites 2 and 3, RS sessions occurred from 3 to 11 times a week, lasted 1 h and consisted of different disciplines (e.g., pony games, jumping, dressage).

### 2.4. Experimental Testing Horses’ Perception of the Environment

#### 2.4.1. Perception of Human: Human–Horse Relationship (HHR) Test

The horses were subjected to standardised human–animal relationship tests in their familiar pasture in group settings [25]. Horses were all tested by the same unfamiliar experimenter (CR, woman, same black clothing for all tests). Two tests were performed, in the same order for all horses:

In the approach contact test (ACT), the experimenter entered the pasture and stood motionless 1.5 m from the horse until it started feeding again. Then, the experimenter approached the horse and attempted to touch its neck. She approached from the side (first left side, second right side), walking slowly and regularly at approximately one step per second, hands hanging down at the sides and looking towards the horse’s shoulder. The horse was free to withdraw from the approach and contact. If the horse threatened the experimenter during her approach, or withdrew from her, she retreated to 1.5 m away from it and renewed the trial. The test was stopped when the experimenter touched the horse’s shoulder continuously for 2 s or after three unsuccessful trials. Both sides of the horse were tested in the same order, i.e., the test was performed twice from the horse’s left first and then from the horse’s right side (2 h interval between changing sides).

The halter fitting test (Halter) was used to test the horses’ reaction towards a human carrying a halter because horses were always fitted with their halter and tied up in the grooming area before a riding lesson. At least half a day after the ACT, the experimenter entered the pasture again, carrying a halter with her left hand. She approached the animal, walking slowly and regularly towards the horse’s left shoulder (halter fitting is commonly always performed on the left side of the horse), at approximately one step per second. When she was near the horse, she stopped walking, put her right arm over the horse’s neck, and fitted the halter.

For both the ACT and Halter tests, all observations were recorded by the experimenter using a digital voice recorder for later transcription. For both tests, the first behaviour of the horses was recorded and five scores were given according to a scale ranging from very “friendly” to very aggressive and to very scared, as follows:

-Score A—the horse looks at the experimenter with ears upright and approaches.-Score B—the horse looks at the experimenter with ears upright and remains where it is.-Score C—the horse shows no evidence of directed attention towards the experimenter (no change in behaviour, no gaze towards the person).-Score D—the horse looks at the experimenter with ears backwards, threatening head posture (head extended) and remains where it is.-Score E—the horse looks at the experimenter with ears backwards and approaches with a threatening posture (neck lowered, head extended or even exposed incisors).-Score F—the horse looks at the experimenter with ears backwards and walks away without threatening posture.-Score G—the horse looks at the experimenter with ears backwards and runs away without a threatening posture.

As in previous studies [33], scores A and B were most often related, as were D and E and F and G. Therefore, horses were considered as having positive (A + B), indifferent (C), aggressive (D + E), or fear (F + G) reactions. The time to touch the horse during the ACT and to fit the halter in the Halter test were scored in seconds by a second experimenter (SAG).

#### 2.4.2. Perception of the Horses’ Environmental Stimuli: Cognitive Judgement Bias Test (JBT)

We used a go/no-go task based on spatial cues from a previous study [28]. Black buckets were used for the test to prevent the horses seeing the contents. The same black buckets were used throughout the habituation, training, and testing phases. At each riding site, the test was conducted in a fenced arena with a sandy substrate familiar to the subjects, but not the same one used for the RS and EAI working sessions and far from disturbances. The habituation, training, and testing phases were videotaped using two JVC EverioR cameras (Dolby, Thailand) placed on a tripod at each side of the arena.

In the arena, a start line was drawn on the sandy ground. The test was carried out by two trained experimenters. Experimenter 1 (E1, EA) prepared the bucket and placed it at a planned location 9 metres from the start line (Figure 1). Experimenter 2 (E2, CR) led the horse into the test arena and positioned it behind the start line. Once E1 had placed the bucket, E2 then released the horse and took a step to the side (Appendix A). The trial was completed when the horse put its nose into the bucket, or after 180 s, which was the maximum latency considered. Once the horse completed the trial, E2 retrieved the horse and brought it back to the start line. E1 measured the time taken by the horse to reach the bucket (from 0 to 180 s). The JBT was divided into 3 consecutive phases: habituation, training, and judgement bias testing.

##### 2.4.2.1. Habituation

Days 1 to 3 comprised the habituation phase with the experimental setting. One session of habituation occurred per day. During this phase, 1 horse at a time was first let free to move from the starting position to only one bucket containing a food reward (pieces of apple, all horses were motivated) placed on one side of the arena (positive location, P). At each site, the position of the reward was on the left for half of the horses and on the right for the other half in order to control for a consistent side bias. The horse was free to approach the bucket. The trial ended either once the horse had approached and put its head into the bucket or after 180 s if the horse did not approach the bucket. In the latter case, the horse was led to the bucket and allowed to eat the mouthful of apple. Each horse was subjected to 5 trials per days.

##### 2.4.2.2. Training

Day 4 to 7 comprised the training phase and depended on each horse’s learning performance. One session of training occurred per day. For the training phase of the negative position (N), a bucket containing apple and vinegar was set on the opposite side of the positive (P) location. There was no visual difference between P and N buckets (i.e., same black colour and size). The 2 positions were presented one at a time (either P or N, no simultaneous bucket); each time the positive location was on the left for half of the horses and on the right for the other half, and the same position was presented twice in a row only once. Thus, a training session included 5 trials and could be P-P-N-N-P or P-N-P-N-P or N-P-P-N-P. Each horse was subjected to the same training position protocol. The training always finished with P (i.e., apple rewarded bucket to maintain horses’ motivation). E1 recorded the latency of each subject to put its nose in the bucket. If the horse did not move towards the bucket in 180 s, E2 led the horse again to the start position and the trial was recorded as completed.

The training ended when the horse learnt the spatial task: left or right related to P or N. The learning criterion was reached if the horses approached all three positive buckets within 20 s (go response) and did not approach either of the two negative buckets within 30 s (no-go response). Each horse was given a learning baseline of at least three training sessions with one training session per day. When needed, additional training sessions were performed until these performance criteria were reached (3 to 7, see results).

##### 2.4.2.3. Testing

The testing phase was performed only once, the day after the completed training. The horses were not all tested on the same day because this depended on each horse’s training performance. During the test, one bucket at a time was presented in 5 positions: P and N and 3 intermediate ambiguous positions, one at a time, with an empty bucket: NN (near-negative), M (middle), and NP (near-positive) (Figure 1). Before the beginning of the test, N and P were presented once each in the arena as a reminder of the training. The test session followed the scheme P-N-NP-P-N-M-P-N-NN-P, where ambiguous locations were preceded alternately by positive and negative locations.

The time required to reach the bucket (latency) and the presence/absence of a go response were recorded for each horse. Go response and short latencies indicated anticipation of a food reward—‘optimistically’ biassed judgement of the ambiguous cue, and high latencies or even failing to approach the ambiguous buckets indicated lower anticipation of food—‘pessimistically’ biassed judgement. To avoid biases caused by differences in baseline walking/trotting speeds due to the size and/or age of individuals, raw latencies recorded to reach the ambiguous positions were transformed into scores, according to Mendl et al. (2010) [34]:$$Adjusted\;latency=\frac{latency\;to\;ambiguous\;location-mean\;latency\;to\;P}{mean\;latency\;to\;N-mean\;latency\;to\;P}$$

This formula returns 0 for P and 1 for N.

### 2.5. Statistical Analyses

The aim of the present study was to test the perception of horses about environmental stimuli, including humans, according to the work they perform. We investigated the influence of the type of work (RS and EAI-RS) and the facility (sites 1, 2 and 3) on the following:

-the perception of humans by horses measured with the proportion of different behavioural responses (from positive to fear responses) and the latency required to touch the horse in the HHR.-the display of pessimist judgement bias measured with the latency required to reach buckets and the presence/absence of a go response in the JBT.

Data were analysed in R v. 4.0.3 [35]. The results were reported as significant at a threshold of *p* ≤ 0.05, and results between 0.05 and 0.08 were reported as ‘trends’ given that these might convey meaningful biological variation [36]. All reported data are presented as a median and Q1–Q3 range. Data were not normally distributed (Shapiro–Wilk test, *p* < 0.001). We fitted separate linear mixed-effects models with the lme4 package [37]. The independence and homogeneity of variances of the models were assessed by inspection of the fitted residuals using the plotresid function in the RVAideMemoire package [38]. Statistical tests (Likelihood Ratio Test) were performed using the ANOVA function with type III sum of squares in the car package [39]. Post hoc analyses were performed using the emmeans function in the emmeans package [40], with Tukey pairwise comparisons. The drop1 function was used for all models to exclude all non-significant factors (except type of work and facility) to reach the minimum adequate model (model selection based on the smaller Akaike Information Criterion, AIC). Sex, age and positive bucket side during training in the JBT were excluded.

#### 2.5.1. Perception of Humans by Horses

The relationships between the behaviour of horses during the ACT-left, ACT-right, and Halter tests were analysed by fitting a separate GLMM, with the proportion of behavioural response within each behavioural category as the dependent variable, the test as a fixed factor, and the horse ID as a random factor to account for repeated tests. As behaviours in the ACT-left and ACT-right were highly positively related, data from both tests were grouped.

The influence of the type of work and facility on the behaviour of the horses during the ACT and Halter tests was analysed by fitting a separate GLM, with the proportion of behavioural response within each behavioural category (i.e., positive, indifferent, aggressive, fear) as the dependent variable and by fitting a separate LM with the latency to touch the horse as the dependent variable. Work (EAI-RS and RS), riding site (sites 1, 2 and 3), and their interactions (work × site) were set as fixed factors.

#### 2.5.2. Perception of Environmental Stimuli by Horses

Separate models were used between the training and the testing phases as the number of training session per horse differed and both contexts included different numbers and locations of presented buckets.

The impact of the type of work and riding site on the horses’ behaviours during the judgement bias training and test was analysed by fitting a separate GLM, with the proportion of go-responses according to the bucket locations as the dependent variable, and an LM for the number of training sessions and the adjusted latency scores to go to the bucket as the dependent variable. Work (EAI-RS and RS), riding site (sites 1, 2 and 3), and their interactions (work × site) were set as fixed factors.

During the judgement bias test, the impact of the bucket location on the proportion of go responses and on the latencies to go to the bucket was analysed by fitting a GLMM and an LMM, respectively. The bucket location was set as a fixed factor and the horse ID as a random factor to account for repeated trials (i.e., P-N-NP-P-N-M-P-N-NN-P).

The relationship between the adjusted latency scores for the three ambiguous bucket locations in the JBT and the total working time (in hours) and EAI working time (in hours and percentage of working time) per week were analysed using Spearman correlation tests.

#### 2.5.3. Relationship Between Horses’ Perception of Humans and Environmental Stimuli

The relationship between the horses’ reaction to humans and judgement bias was analysed using a separate LM, with the adjusted latency scores for the three ambiguous bucket locations during the judgement bias test as the dependent variables and the behavioural score during the ACT and the Halter tests, work (EAI-RS and RS), riding site (site 1, 2 and 3), and their interactions (work × site) set as fixed factors.

## 3. Results

### 3.1. Perception of Human by Horses: Human–Horse Relationship Test (HHR)

During the approach contact test (ACT left and right), 41% of horses showed positive behaviours, 20% were indifferent, 22% showed aggressive reactions, and 17% fear behaviours (Figure 2). The proportion of positive behaviours differed according to the riding site (Table 2), but not according to the type of the work (GLM, N = 30, X^2^ = 20.55, *p* < 0.001; X^2^ = 0.00, *p* = 0.99, respectively, Table 2). More horses from sites 1 and 3 showed positive behaviours compared to site 2 (site 1 vs. site 2: t = 5.28, *p* < 0.001; site 1 vs. site 3: t = 2.04, *p* = 0.12, site 2 vs. site 3: t = −2.28, *p* = 0.07). The proportion of indifferent behaviours differed according to the site and there was a significant interaction between site and the type of work (X^2^ = 17.69, *p* < 0.001; X^2^ = 4.70, *p* = 0.02, respectively, Table 2). More horses from site 2 showed indifferent behaviours compared to sites 1 and 3 (site 1 vs. site 2: t = −5.17, *p* < 0.001; site 1 vs. site 3: t = 0.35, *p* = 0.93, site 2 vs. site 3: t = 4.49, *p* < 0.01). More EAI-RS horses from site 2 showed indifferent behaviours (site 2 EAI-RS vs. RS: t = −3.39, *p* = 0.03). The proportion of aggressive behaviours did not differ according to the riding site or type of work (Table 2). There was a significant interaction between site and the type of work for the proportion of fear behaviours (X^2^ = 9.67, *p* < 0.01), but post hoc comparisons did not show significant differences (*p* > 0.05 for all). Latencies to touch the horses did not differ according to the riding site or type of work (LM, F = 0.66, *p* = 0.53; F = 0.29, *p* = 0.59, respectively, Appendix A).

During the Halter test, 43% of horses showed positive behaviours, 27% were indifferent, 13% showed aggressive behaviours, and 17% fear behaviours. There was no relationship between behaviours in the ACT-left and -right and the Halter tests (GLMM, *p* > 0.05). The proportion of positive behaviours differed according to the riding site but not the type of work (GLM, N = 30, X^2^ = 21.90, *p* < 0.001; X^2^ = 0.61, *p* = 0.43, respectively, Table 2). More horses from site 1 showed positive behaviours compared to sites 2 and 3 (site 1 vs. site 2: *p* < 0.001; site 1 vs. site 3: *p* = 0.02, site 2 vs. site 3: *p* = 0.15). The proportion of indifferent and aggressive behaviours tended to differ according to the riding site but not the type of work (Table 2), but post hoc comparisons did not show significant differences (*p* > 0.05 for all). The proportion of fear behaviours differed according to the riding site but not the type of work (X^2^ = 6.20, *p* = 0.04; X^2^ = 0.35, *p* = 0.55, respectively). Post hoc comparisons did not show significant differences according to the riding site (0.21 < *p* < 1). There was a significant interaction between the riding site and type of work in regard to the latencies to touch the horse (LM, F = 6.30, *p* = 0.02, Appendix A). RS horses from site 2 tended to require more time to be touched compared with RS horses from site 1 (t = −3.03, *p* = 0.07).

### 3.2. Perception of Horses’ Environmental Stimuli: Cognitive Judgement Bias Test (JBT)

#### 3.2.1. Training

Three to seven training sessions were required for horses to reach the learning criterion in the go/no-go spatial discrimination task. The number of training sessions required differed significantly between the three riding sites (LM, F = 14.92, *p* = 0.009, Table 3) and there was a tendency for an interaction with the type of work (F = 3.10, *p* = 0.06). EAI-RS horses from site 3 needed more training sessions compared to horses from site 1 (both EAI-RS and RS, t = −3.33, *p* = 0.03; t = −4.58, *p* < 0.001) and to EAI-RS horses from site 2 (t = −3.25, *p* = 0.03). At the end of the training, mean latencies to reach the positive location averaged 9.73 ± 0.66 s. There was a significant difference between sites (F = 4.07, *p* = 0.03, Table 3). Horses from site 3 had higher latencies to reach the positive location compared with horses from site 2 but not site 1 (site 1 vs. site 2: t = 1.13, *p* = 0.50: site 1 vs. site 3: t = −1.77, *p* = 0.20; site 2 vs. site 3: t = −2.61, *p* = 0.04). Mean latencies to reach the negative location did not differ significantly between sites and the type of work (all *p* > 0.05; Table 3).

#### 3.2.2. Testing

During the judgement bias test, the responses of horses to the positive and negative locations were similar to those during training (i.e., large proportions of go responses and short latencies to approach the positive location, and low proportions of go responses and longer latencies to approach the negative location). Bucket location affected the proportion of go responses (GLMM, *p* < 0.05 in all cases, Figure 3) and latencies (LMM, X^2^ = 820.11, *p* < 0.0001, Appendix A). Overall, 100% of horses approached the positive (P) bucket at least once, 100% the nearest positive (NP), 97% the middle (M) (94% EAI-RS; 100% RS), 77% the nearest negative (NN) (75% EAI-RS; 79% RS), and 33% the negative (N) bucket (37% EAI-RS; 29% RS).

There was no significant influence of riding site and the type of work on the proportion of go responses and the adjusted latency scores to the ambiguous NP bucket (Table 3). For the ambiguous M bucket, there was no significant effect on the proportion of go responses, but the site tended to influence the adjusted latency (LM, F = 3.02, *p* = 0.07) and there was a significant influence of the type of work (F = 4.58, *p* = 0.04, Figure 3). EAI-RS horses from site 2 tended to have a higher adjusted latency to go to the M ambiguous location (t = 3.01, *p* = 0.06). For the NN ambiguous bucket, there was a significant interaction between the site and the type of work for the proportion of go responses (GLM: X^2^ = 8.59, *p* = 0.01), but post hoc comparisons were not significant. There was a significant influence of the riding site on the adjusted score to the NN bucket (LM, site: F = 8.78, *p* = 0.01), but post hoc comparisons were not significant. Overall, there was no significant relationship between either total working time, or EAI working time/percentage of working time, and adjusted latency scores to the three ambiguous buckets (Spearman test, *p* > 0.05 for all, Appendix A).

### 3.3. Relationship Between HHR and JBT

Overall, the score per horse during the Halter tests did not predict adjusted scores for the three ambiguous bucket locations during the judgement bias test (Appendix A). The score per horse during the ACT tended to predict adjusted latency for the NN bucket (LM, F = 2.70, *p* = 0.07). Horses that showed indifferent behaviours tended to show a higher adjusted latency score for NN buckets (t = 2.70, *p* = 0.07).

## 4. Discussion

The aim of this study was to test the perception of horses of their environmental stimuli, including humans, according to the work they are involved in. As an indicator of compromised welfare, we expected that horses involved in both riding school lessons and equine-assisted interventions would show a pessimistic bias during a cognitive judgement bias test (JBT) and would be less interactive during human–horse relationship tests (HHR) than horses only involved in riding school lessons. We also predicted that facility management practises inducing compromised horse welfare may lead to pessimistic bias in the JBT and less positive behaviours in the HHR. In contrast, we found no clear significant effect of the type of work on the expression of a pessimistic bias and behaviours towards humans. Testing a larger sample of animals might have led to different findings, and further studies are needed. However, we did observe a modulating effect through the interaction between work and facility. Overall, the facility management protocols in our study sites appeared to influence the horses’ perception of their environment compared to their type of work.

### 4.1. Population Perception of Environmental Stimuli, Including Humans

Our findings revealed an overall positive perception of the environmental stimuli in our study population in the JBT. Most of the horses anticipated a positive outcome with a go response to the NP and M ambiguous location. In particular, 77% of horses anticipated a positive outcome in the nearest negative (NN) location, suggesting an optimistic judgement bias. The NN location is the more discriminant in highlighting presence/absence of pessimistic bias, e.g., [28], and further investigation is needed on the characteristics of the 23% of horses anticipating a negative outcome in the NN location. Our results could be related to some extent to limitations of the JBT, such as that ambiguity could be perceived as novel rather than intermediate (mixed positive/negative) stimuli [41,42], and/or that discrimination training could act as cognitive enrichment for animals, hence influencing affect and responses during testing [43].

Our results could also be explained by a relative positive mood in the tested horses. However, this was not confirmed by our results in the human–horse relationship tests. In the HHR test, around 40% of horses displayed positive behaviours (ACT 41%; Halter test: 43%) and 60% of horses displayed indifferent and negative coping behaviours. The proportion of horses displaying signs of aggression was around 20% for each test. Part of the horses may have associated humans with negative experiences such as discomfort during handling or riding. Working conditions during riding school lessons, in particular the potential emotional tensions induced during these lessons, the rider’s posture and actions, and/or the quality or fitting of the working equipment, could increase lameness, dorsal problems, and other musculoskeletal pain [11,44], possibly related to aggressive reactions in human–horse relationship tests [33]. Further investigation is needed on the characteristics of the 17% of horses showing fear behaviours suggesting horses’ perception of humans as a threat and eliciting avoidance. Fear behaviours towards humans have been related to the influence of invasive early experiences and daily interaction [25]. Signs of indifference were around 20% for each test. This result raised questions about the meaning of a form of unresponsiveness. Indifferent behaviours to a human’s approach have been interpreted as neutral behaviour [45], or related to a depressed state [46], or reported as a state before change into negative behaviours as the human approach becomes more invasive [47]. Indifferent behaviour may thus be related to a more negative than positive perception of humans by horses and/or an overall negative mood, which was confirmed by the positive relationship between indifferent behaviours in the HHR and higher latencies to ambiguous NN bucket locations in the JBT.

### 4.2. Facility Influence

Facility significantly impacted the horses’ perception of their environment, including humans. Horses from site 1 and site 3 seemed optimistic in the JBT and positive towards humans in the ACT/Halter test, whereas horses from site 2 showed higher latencies to reach the ambiguous middle location in the JBT, showed more indifferent behaviours in the ACT, and greater fear behaviours in the Halter test. The overall results could imply variation in factors between facilities, such as environmental conditions during tests or riding site differences that we could not measure [48,49]. This could also be related to management of the living and riding conditions of the horses. Several studies have highlighted the relationship between ad libitum feeding access and general good welfare in horses (e.g., [50]). An epidemiological study showed that access to roughage was the primary factor predicting good welfare in horses, with riding parameters being the second most influential factor [51]. When facilities kept their horses outdoors, with a high roughage/low pellet regimen, and used groundwork or unbitted riding, the horses showed fewer indicators of compromised welfare such as aggressive behaviours in human–horse tests or abnormal repetitive behaviours [14].

### 4.3. Type of Work Influence

The lack of an effect of the type of work on the expression of a pessimistic bias/negative perception of humans could have been masked by several factors such as the facility location, management (i.e., overall housing and feeding), equestrian “culture” related to working conditions, and the quality of human–horse interaction. This is confirmed by significant interactions between the type of work and facility and the horses’ behaviour in the JBT and HHR tests. In addition to EAI working activities, horses may also experience living conditions, human–horse interactions, and other working activities that influence their welfare. This can be explained by differences in the quantity and/or quality of the working conditions between facilities. Variability in workload quantity between sites could be related to managers’ horse choice criteria according to the intended activity. RS managers reported personality and physical characteristics as the main criteria, and EAI managers put more emphasis on horses with a positive relationship with humans [14]. This suggests that the same horse could be often chosen and, hence, have a high workload that may influence its welfare.

However, the present study did not highlight a relationship between workload quantity or proportion of EAI activity and behaviours in the JBT, suggesting an influence of the quality of working conditions. Horses at riding site 1 were mostly involved in EAI sessions that lasted an hour, with a low amount of time spent in grooming; mounting was mostly associated with a lift and riding teachers promoted lengthened rein actions. Horses at riding sites 2 and 3 were involved in RS activities mainly, with more restrictive techniques (e.g., short and tensed reins, reining, personal observation, which has been related to high and hollow neck postures and back disorders in horses [11]). EAI sessions lasted 2 h, more time was dedicated to grooming, and mounting was not associated with a lift. Previous studies have shown an increase in “stress”-related behaviour during grooming, mounting, and dismounting phases in EAI and RS sessions, compared to the riding *per se* [52]. The authors suggested that attention must be given to the riding equipment, handlers, and the riders’ behaviour, which may cause discomfort during these particular grooming and mounting–dismounting phases of the riding session. A recent study also showed higher tactile reactivity (tested experimentally) in EAI horses compared to RS and EAI-RS horses. This difference could be related to human actions during grooming phases, as EAI beneficiaries mostly brushed the hindquarters and showed more fragmented actions (grooming stops with breaking contact between the brush and the horse’s body), which may cause the horse discomfort [53]. Finally, not only riders but the presence of other humans, such as handlers or therapists, involved in EAI sessions could induce stress in horses [54]. Further studies are needed to disentangle the welfare outcomes of human actions (e.g., handlers, riders, riding teachers) during different phases of EAI sessions and also of management during the riding phase (including equipment choice and use).

### 4.4. Limitations of the Study

Our investigation of horses’ perception of their environment, including of humans, was necessarily indirect, and we should acknowledge that variations in individual coping abilities may influence their emotional state. We used the JBT and HHR approaches because previous studies suggested a chronic effect of EAI activity to result in a lack of attention and interaction towards humans (e.g., [26]). The JBT and HHR test may have revealed whether EAI horses showed altered/modified underlying cognitive mechanisms such as perception and categorization [5]. It would be interesting to test for other potential cognitive biases (e.g., attention bias [48]) or other cognitive functions. For example, it would be important to test inhibitory control functions according to the type of work because EAI horses may often inhibit behavioural expressions associated with emotions, which may impact their welfare. Behavioural inhibition may be related to the training of the horses or to their personality, and inappropriate working/training conditions could also influence their personality [55]. The behaviours exhibited may also be influenced by an individual’s coping strategy; the coping style may be either more active (i.e., manifesting avoidance or escape behaviours, or defence behaviours) or more passive (i.e., manifesting immobility or displacement behaviours).

Further studies are needed to link horse behaviours in the JBT and HHR test with other behavioural welfare indicators which could highlight passive coping styles. Time budget observations may reveal some indicators of compromised welfare, such as “depressed/apathetic” states, stereotypic/abnormal repetitive behaviours, and positive emotion indicators (e.g., snorts [56]). The next step would be to relate the horse’s behaviours in their daily life, in the JBT, and in the HHR with their behaviour during EAI and RS sessions in order to explore the link between acute emotional expressions at work and chronic emotional state. In particular, previous studies focusing on horse behaviour at work showed contrasting results (i.e., more, less, no difference in discomfort behaviour frequencies between EAI and RS sessions, e.g., [20,24,52,57]). Given the high prevalence of back and orthopaedic pain in riding school horses, it would be interesting to add a measurement of horses’ neck posture [11] or to assess the presence of pain, which is also related to cognitive alteration in horses [58]. It would be important to compare horses involved only in EAI to control for confounding factors such as personality and the history of horses. However, a recent study suggested that horses working in both EAI and RS were more at risk of compromised welfare due to either the emotional/physical strain of EAI in addition to the daily riding lessons, or to the same management of riding teachers in EAI and conventional riding lessons; notably, EAI riders may have disabilities that increase the horses discomfort at work [14].

## 5. Conclusions

This study considered equine perceptions of environmental stimuli, which is of great importance for assessing horses’ long-term welfare. We propose that the predicted emotional/physical strain of EAI added to the daily riding lessons may have been masked by the overall facility management. This is confirmed by some significant interactions between the type of work and facility influence on horses’ behaviours in the JBT and HHR test. Changes in working techniques, such as the use of positive training (e.g., positive reinforcement), may mitigate the proportion of horses displaying aggressive or fear behaviours in the HHR test and showing pessimism in the JBT. Testing a larger sample of animals might have led to different findings, and further studies relating cognitive functions, chronic emotional state, and acute emotion expressions in EAI sessions are needed. This integrative framework will further elucidate whether there may be some aspects of EAI that are more difficult to manage for horses than others.

## Figures and Tables

**Figure 1 animals-15-00607-f001:**
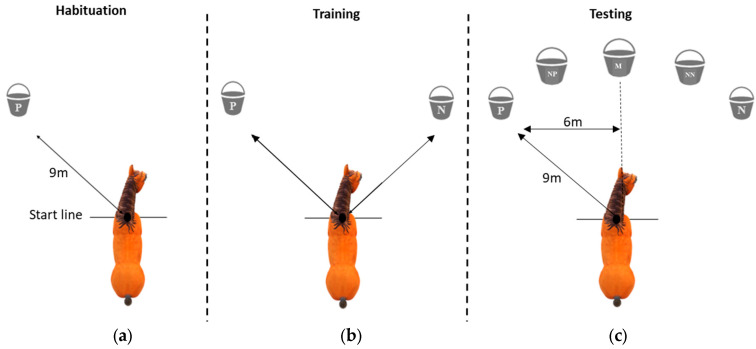
Diagram of the arena showing (**a**) the JBT habituation phase with only one bucket containing a food reward placed on one side of the arena (positive location, P, on the left for half of the horses and on the right for the other half); (**b**) the JBT training phase based on a spatial cue (left or right) with only one bucket at a time either at the positive location, P, or at the negative location, N; (**c**) the JBT testing phase with only one bucket at a time with five possible positions of the bucket [28]: positive (P), nearest to positive (NP), middle (M), nearest to negative (NN), and negative (N). Positive and negative bucket positions were 12 m apart. All bucket positions, during both training and testing, were 9 m from the start position of the horse.

**Figure 2 animals-15-00607-f002:**
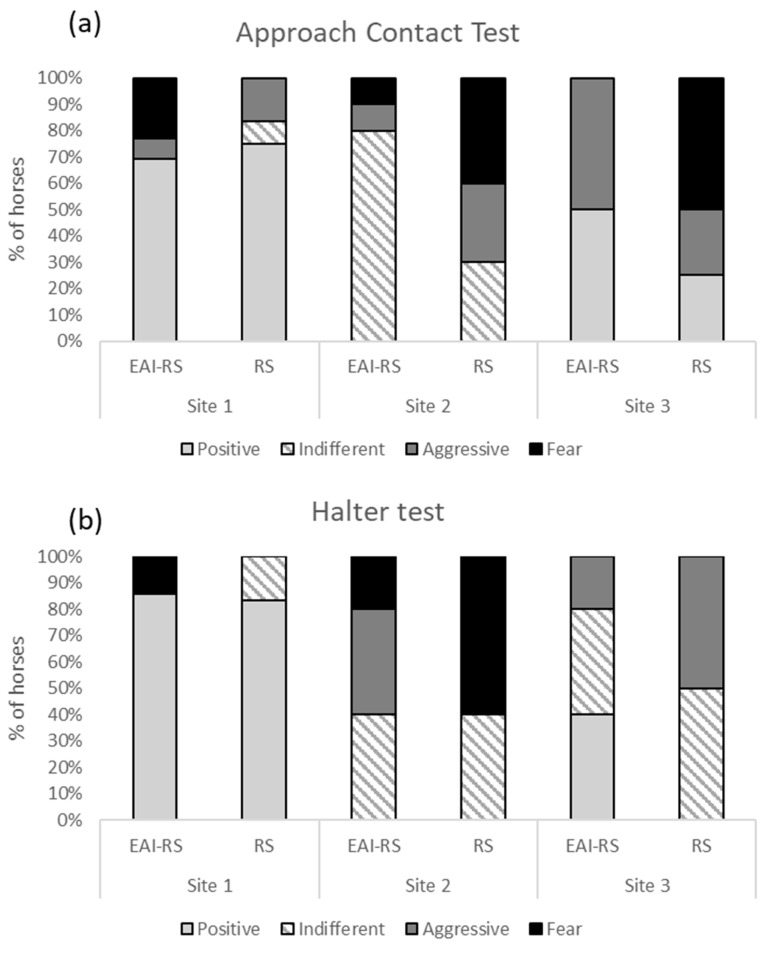
Proportion of horses displaying positive, indifferent, aggressive, or fear behaviours during (**a**) the approach contact test (left and right sides of the horse); (**b**) the Halter test. Horses were from 3 different riding facilities (site 1, 2, and 3). EAI-RS: horses involved in equine-assisted interventions and riding school (N = 16); RS: horses involved only in riding school (N = 14).

**Figure 3 animals-15-00607-f003:**
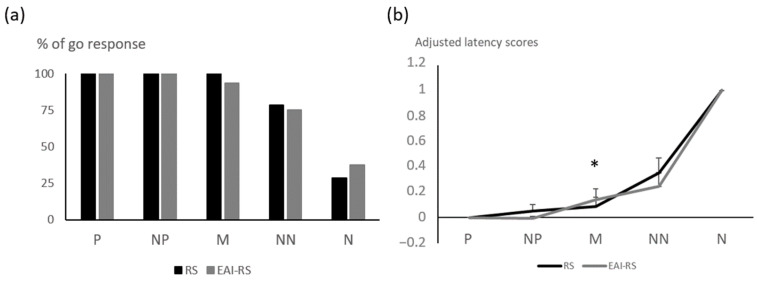
(**a**) Proportion of horses approaching the positive (P) and negative (N) learnt locations and the three ambiguous locations: near positive (NP), middle (M), and near negative (NN); (**b**) adjusted latency scores for the judgement bias test. RS: riding school horses (N = 14), EAI-RS: equine-assisted intervention + riding school horses (N = 16). GLMM and LMM * *p* < 0.05.

**Table 1 animals-15-00607-t001:** Animals’ characteristics, housing, and working conditions according to the riding facilities.

			Riding Facility		
			Site 1	Site 2	Site 3
			13	10	7
Horses’ characteristics	Activity	EAI-RS *	6	5	5
		RS *	7	5	2
	Sex	Geldings	9	6	5
		Mares	4	4	2
	Type	Horses	6	0	0
		Ponies	7	10	7
	Breed	Unregistered	62%	100%	57%
		French pony	0%	0%	43%
		Fjord pony	23%	0%	0%
	Age (mean ± SEM y.o.)		15.8 ± 1.2	17.1 ± 1.7	15.4 ± 2.1
Feeding	Pellets	Nb * of meals/day	0	1	3
		Quantity (L/day)	0	1.5	1.5
	Roughage	Nb * of meals/day	Ad libitum	1	Ad libitum
		Quantity (Kg/j)	Ad libitum	3	Ad libitum
	Grass		x *	x	Ø *
Housing	Type	Paired Stall (3.5 × 3.5 m)			x
		Group stall (6 × 3 m)		x	
		Paddock hay (0.5 m^2^)	x		x
		Paddock Ø hay (0.5 m^2^)		x	
		Pasture (>1 ha)	x	x	
	Time per week	Pair stall	Ø	Ø	57%
		Group stall	Ø	10%	Ø
		Paddock	10%	30%	43%
		Pasture	90%	60%	Ø
Working	Number hour per week [range]	RS	[2 h 45–7 h 45]	[8–11 h]	[3–10 h]
		EAI	[1–5 h]	2 h	2 h
	% of EAI [range]	EAI-RS	[28–86%]	[9–11%]	[10–17%]
	Day off per week		2	2	2

* EAI-RS: horses involved in equine-assisted interventions and riding school; RS: horses involved only in riding school, x: present; Ø: absent, Nb: number.

**Table 2 animals-15-00607-t002:** The influence of the riding facility (site 1, 2 and 3), the type of work (riding school lessons and equine-assisted intervention + riding school lessons), and their interaction on the proportion of positive, indifferent, aggressive, or fear behaviours during the approach contact (ACT) and the Halter tests (N = 30).

Tests				
		X^2^	Df	*p*
ACT ^1^	Positive behaviour			
	Site	20.55	2	**≤0.001**
	Work	0.00	1	0.99
	Site × work	0.57	2	0.75
	Indifferent behaviour			
	Site	22.64	2	**≤0.001**
	Work	8.46	1	**≤0.01**
	Site × work	0.00	2	1.00
	Aggressive behaviour			
	Site	3.99	2	0.13
	Work	0.27	1	0.60
	Site × work	1.75	2	0.42
	Fear behaviour			
	Site	1.01	2	0.58
	Work	0.99	1	0.32
	Site × work	9.66	2	**≤0.01**
Halter	Positive behaviour			
	Site	21.90	2	**≤0.001**
	Work	0.61	1	0.43
	Site × work	1.05	2	0.59
	Indifferent behaviour			
	Site	4.92	2	*0.08*
	Work	0.39	1	0.53
	Site × work	1.31	2	0.52
	Aggressive behaviour			
	Site	5.00	2	*0.08*
	Work	0.49	1	0.48
	Site × work	3.39	2	0.18
	Fear behaviour			
	Site	6.20	2	**0.04**
	Work	0.35	1	0.55
	Site × work	2.68	2	0.26

^1^ ACT: approach contact test (left and right combined); the Likelihood Ratio Test statistics (X^2^), the denominator degrees of freedom (Df), and *p* values (GLMM, **bold values**
*p* < 0.05; *italic values* 0.05 < *p* < 0.08) are shown.

**Table 3 animals-15-00607-t003:** Influence of the riding site (site 1, 2 and 3), the type of work (riding school lessons and equine-assisted intervention + riding school lessons), and their interaction on the performances during training to the spatial discrimination task and during the judgement bias test (JBT).

	F	Df	*p*
Training: number of sessions			
Site	7.01	2	**≤** **0.01**
Work	1.42	1	0.24
Site × work	3.10	2	*0.06*
Training: mean latency to positive location			
Site	4.07	2	**0.03**
Work	0.19	1	0.66
Site × work	0.06	2	0.94
Training: mean latency to negative location			
Site	1.56	2	0.23
Work	0.21	1	0.65
Site × work	0.27	2	0.77
JBT: NP adjusted latency			
Site	2.02	2	0.15
Work	0.43	1	0.51
Site × work	0.37	2	0.69
JBT: M adjusted latency			
Site	3.02	2	*0.07*
Work	4.58	1	**0.04**
Site × work	1.57	2	0.23
JBT: NN adjusted latency			
Site	8.78	2	**0.01**
Work	2.08	1	0.16
Site × work	1.61	2	0.22
	**X^2^**	**Df**	** *p* **
JBT: NP go responses			
Site	0	2	1
Work	0	1	1
Site × work	0	2	1
JBT: M go responses			
Site	2.48	2	0.29
Work	1.49	1	0.22
Site × work	0	2	1
JBT: NN go responses			
Site	0.66	2	0.72
Work	0.13	1	0.72
Site × work	8.59	2	**0.01**

JBT: judgement bias test; three ambiguous locations: near positive (NP), middle (M), and near negative (NN); the Likelihood Ratio Test statistics (F and X^2^), the denominator degrees of freedom (Df), and *p* values (LMM and GLMM, **bold values**
*p* < 0.05; *italic values* 0.05 < *p* < 0.08) are shown.

## Data Availability

The raw data generated and analysed during the current study are available upon request to the corresponding author.

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
