# Peer review of "Through a Horse’s Eyes: Investigating Cognitive Bias and Responses to Humans in Equine-Assisted Interventions"

_animals, 2025, doi:10.3390/ani15040607_

Round 1
Reviewer 1 Report
Comments and Suggestions for Authors
This is an interesting multisite which investigated the effects of behavioural tests on horses which were used either purely for Riding School work or also for equine assisted interventions. It appears that management may be a more influential factor than use alone.
Some of the numbers are very confusing. I will highlight this in my more specific comments.
There is evidence from a number of different countries that a large proportion of riding school horses are lame. Chronic discomfort can influence behaviour. Many of these horses go unrecognised.
I think that a limitation of this study is that the horses did not undergo independent orthopaedic assessment (including ridden exercise).
I would suggest that the impact of chronic musculoskeletal pain on behaviour should be discussed.
I suggest that the inclusion of the results in the text, in tables & in figures creates duplication.
There are numerous small grammatical errors.
Lay person summary
Line 19 more negative than what?
Abstract
Line 31 do you mean horses' perceptions of humans?
Line 35 ‘Our 33 hypothesis was that horses engaged in both types of work would experience a more negative impact, as both activities may be demanding’ - A more negative impact compared with what?
EAI is not necessarily physically demanding - it is often very slow & of relatively short duration - but there are potential problems such as ? asymmetrical weight distribution? rider size? saddle fit for rider?
Were the horses pain-free?
Introduction – quite long – but provides some good background information
Methods
Lines 173, 179, 185 Numbers don't add up or agree with Table 1
- site 1, n=13, site 2 n=10, site 3 n=7 Total = 30
Line 176 Riding sites (not site)
line 187 ‘…. either in pair stalls 187 with straw bedding or? rather than either
All Table and figure legends need to be able to be read independently from text so more details are needed
Table 1 give SE to one decimal place
Were data normally distributed ? If yes give mean & SD, it not median & IQR and range
Number of hours per week - Why are some numbers in brackets - ? because it is a range, not absolute number?
Table confusing - were there any horses only involved in RS? Does not seem consistent with text – section 2.3.
Line 201 All text should be consistent in tense to this should be in past tense
‘worked’
Line 203 of, not in ‘..consisted mostly of grooming…..’
Line 205 'and an important part of work related to EAI activities ' is redundant. Information is provided in the next sentence.
Line 208 of not on ‘…. consisted of 5 minutes…..’
Line 220 of not on ‘…. consisted of 5 minutes…..’
Line 268 walks away (not walk away)
Line 270 runs away (not run away)
Line 278 It would be clearer of the text was amended to ‘…from a previous study [28].’
line 284 planned (not planed)
Line 286 I suggest that this is amended to ‘….then E2 released the horse and took a step...’
line 332 cameras (not camera)
Results
Line 394 sites (not site)
Line 430 I suggest that this is amended to '...touched compared with horses from site 1'
Do we really need same results in text, table& figures?
I like the figures – they depict the information well.
Line 456 ‘…location compared with horses…’
Discussion
The Discussion is relative long. Try to avoid repetition of results.
The relatively high proportion of horses showing signs consistent with fear and/ or aggression is worthy of comment.
What can be done about this?
Was your sample size big enough to show differences between RS & RS+EAI?
Please highlight the limitations of the study.
Line 534 - less than what?
Line 610 Did you actually observe 'more constraining riding techniques (e.g. short and tensed reins applying repeated tension on the horse’s mouth ' or is this just an assumption?
Supplementary material
P values to 2 decimal places is enough. Please see previous comments about normality of results or otherwise
All abbreviations in the tables need to be defined
The Table legends should have sufficient detail to that they can be read independently from the text
Comments on the Quality of English Language
Suggestions made for improvement
Author Response
This is an interesting multisite which investigated the effects of behavioural tests on horses which were used either purely for Riding School work or also for equine assisted interventions. It appears that management may be a more influential factor than use alone.
Some of the numbers are very confusing. I will highlight this in my more specific comments.
There is evidence from a number of different countries that a large proportion of riding school horses are lame. Chronic discomfort can influence behaviour. Many of these horses go unrecognised.
I think that a limitation of this study is that the horses did not undergo independent orthopaedic assessment (including ridden exercise).
I would suggest that the impact of chronic musculoskeletal pain on behaviour should be discussed.
I suggest that the inclusion of the results in the text, in tables & in figures creates duplication.
There are numerous small grammatical errors.
Thank you for your very valuable comments that improved the manuscript. We checked the English and grammatical errors throughout.
Lay person summary
Line 19 more negative than what?
L19: More negative than RS horses
Abstract
Line 31 do you mean horses' perceptions of humans?
L31: yes, perception of humans: “We investigated the cognitive judgment bias (pessimistic vs optimistic) and the perception of humans (negative vs positive) in horses from three different facilities”
Line 35 ‘Our hypothesis was that horses engaged in both types of work would experience a more negative impact, as both activities may be demanding’ - A more negative impact compared with what?
L34: We hypothesised that horses engaged in both types of work would be more negatively impacted than RS horses, because the two activities may be demanding.
EAI is not necessarily physically demanding - it is often very slow & of relatively short duration - but there are potential problems such as ? asymmetrical weight distribution? rider size? saddle fit for rider?
The potential problems are unusual behaviours and posture leading to asymmetrical weight distribution. To our knowledge, there is no previous study on rider size or saddle fit influence during EAI work. We added the influence of asymmetrical weight distribution in the introduction (Line 90).
Were the horses pain-free?
We do not have access to these data. Animal husbandry and care were under the management of the riding school staff. The riding school managers provided the authors with the information of osteopathic exams for each horse once a year.
Introduction – quite long – but provides some good background information
Thanks
Methods
Lines 173, 179, 185 Numbers don't add up or agree with Table 1
- site 1, n=13, site 2 n=10, site 3 n=7 Total = 30
Yes, the information lines 173, 179 and 185 were about the number of individuals in the herd. Within each herd, we tested a subset of horses involved in EAI-RS and RS in the same way.
We added the detailed “individuals in the herd” lines 169, 176 and 182.
Line 176 Riding sites (not site)
L172: done
line 187 ‘…. either in pair stalls 187 with straw bedding or? rather than either
L178: English corrected
They were housed either in a paddock with hay available ad libitum (43% of time), either in pair stalls with straw bedding and hay provided twice per day (57% of time).
All Table and figure legends need to be able to be read independently from text so more details are needed
Legends have been changed throughout the document
Table 1 give SE to one decimal place
Done
Were data normally distributed ? If yes give mean & SD, it not median & IQR and range
Data were not normally distributed. Thus, mixed models were used and independence and homogeneity of variances of the mixed models were assessed by inspection of fitted residuals using the plotresid function. Data are given in median & IQR range.
Number of hours per week - Why are some numbers in brackets - ? because it is a range, not absolute number?
Numbers in brackets are the range. We added detailed in Table 1.
Table confusing - were there any horses only involved in RS? Does not seem consistent with text – section 2.3.
There were 14 horses only involved in RS. Corrected in the text L194-196:
(RS, 6 mares, 8 geldings, X=13.4±0.7 y.o) and 16 horses were involved in both RS and EAI activities (EAI-RS, 4 mares, 12 geldings, X=18.2±2.3 y.o, see Table 1).
Line 201 All text should be consistent in tense to this should be in past tense
Done (L194-225).
Line 203 of, not in ‘..consisted mostly of grooming…..’
Done (L201).
Line 205 'and an important part of work related to EAI activities ' is redundant. Information is provided in the next sentence.
The entire paragraph has been re-written
Line 208 of not on ‘…. consisted of 5 minutes…..’
Done (L210).
Line 220 of not on ‘…. consisted of 5 minutes…..’
Done (L216).
Line 268 walks away (not walk away)
Done (L270).
Line 270 runs away (not run away)
Done (L272).
Line 278 It would be clearer of the text was amended to ‘…from a previous study [28].’
Done (L282): We used a go/no-go task based on spatial cues from a previous study [28].
line 284 planned (not planed)
Done (L291)
Line 286 I suggest that this is amended to ‘….then E2 released the horse and took a step...’
Done (L293) “Once E1 had placed the bucket, then E2 released the horse and took a step to the side”.
line 332 cameras (not camera)
Done (L287).
Results
Line 394 sites (not site)
Done (L395)
Line 430 I suggest that this is amended to '...touched compared with horses from site 1'
It has been rephrased (L 429) RS horses from site 2 tended to require more time to be touched compared with RS horses from site 1
Do we really need same results in text, table& figures?
I like the figures – they depict the information well.
Line 456 ‘…location compared with horses…’
Done (L456)
Discussion
The Discussion is relative long. Try to avoid repetition of results.
We removed repetition of the results.
The relatively high proportion of horses showing signs consistent with fear and/ or aggression is worthy of comment.
Thanks for the advice, see lines 640-657
What can be done about this?
Sorry, as I do not know which sentence is commented.
Was your sample size big enough to show differences between RS & RS+EAI?
Testing a larger sample of animals might have led to different findings and further studies are needed (line 621)
Please highlight the limitations of the study.
Limitations are highlighted in the last part of the discussion
Line 534 - less than what?
Done (L617): than horses only involved in riding school lessons
Line 610 Did you actually observe 'more constraining riding techniques (e.g. short and tensed reins applying repeated tension on the horse’s mouth ' or is this just an assumption?
I actually observe more constraining riding techniques and equipment in ridinf sites 2 and 3 compare to riding site 1.
L 694: more restrictive techniques (e.g. short and tensed reins, reining).
Supplementary material
P values to 2 decimal places is enough. Please see previous comments about normality of results or otherwise
Done throughout the manuscript
All abbreviations in the tables need to be defined
Done
The Table legends should have sufficient detail to that they can be read independently from the text
Legends have been changed.
Reviewer 2 Report
Comments and Suggestions for Authors
This is an excellent concept and likely also an excellent scientific study. The lack of clarity in the methods and certain sections of writing can be addressed to make a great publication. The authors should truly be applauded for addressing the mental domain of equine welfare in relation to type of work.
There is significant need for improvement of how the JBT methods are described, as such it is so unclear that it inhibits interpretation of methodology. Because of this it is hard to truly evaluate the statistical approaches, however, there is some question within their modelling if combined models/covariates were appropriately included. The authors need to address the confounding of utilizing testing restricted to evaluation of active coping methods, as well as interaction between site and quality of the human-animal interactions in addition to their focus on environmental/nutritional management being a driver of site differences. These challenges with the paper should be addressed before publication to ensure the paper conveys their hard work and very pertinent approach to an area of equine behaviour and welfare that much needs this type of support and investigation.
In addition to in-line comments in the pdf, there are a few general comments here:
- Lack of clarify on JBT testing leads to inability to truly assess scientific rigor and appropriateness of subsequent sections (analysis, results, discussion)
- Time budget methods and analysis needs to be included in methods/results not as a small mention in discussion
Introduction
- Sentence grammar in introduction needs attention
- Introduction should be re-written for clarity of concepts and relation between the concepts presented and the aims of the study. As is, it jumps around a lot and is a bit hard to follow. There is a LOT of great information within it and would love to see that come through for the readers.
- Are they measuring compromised state related to apathy? (see intro last paragraph in predictions)
Methods
- 3 sites – site 1 large EAI load
- Description of sites/work needs clarification. For example, EAI seems very similar between all sites, and similar for RS, however, how it is presented is not similar and makes the reader have to think/question where the differences are (e.g. in horse handling/tacking for RS. One consideration maybe to re-write relative to outcomes of focus being together (e.g. EAI described in general then differences by site, then RS in general and differences by site, then types of riders) rather than current approach of describing by site. Not a necessary edit, but something to reflect on when reviewing how this section appears to support a reader connecting with your methods
- Why was no assessment of individual personality performed?
- HHR
o Scores are biased towards active coping styles and terminology is not representative of exclusivity. Suggest renaming “negative” scores of D +E to “defensive” (which doesn’t necessarily capture the ambiguity of ear position in D) or maintain consistency with results section of “aggressive”. However, standing with ears back and not engaging is not a exclusive sign of aggressive behaviour in horses, so proceed with labelling scores with caution as to the implications and assumptions for the readers
- JBT – needs major revisions for clarity of methods
o Description of training phase needs clarity of bucket pattern differing between horses or within the same horse’s training, P/N bucket differentiation relative to sensory systems of the horse.
- Analysis
o Add section headings within analysis, and a statement of the purpose of each analysis (e.g. “To investigate XXX prediction/hypothesis, a XXX model was used) prior to details of the models
o needs clarification, which relates to needs in methods section.
o What is the purpose and use of the adjusted latency scores?
o ACT
§ Please provide assessment of relationship between left/right ACT
§ consider a model comparing positive vs. negative valanced reactions as the lack of score differences with current approach to combining scores questions if the chosen “grouping” of scores is reflective of the underlying intention of “positive vs. negative” reactions, particularly with the non-exclusivity of behaviours in separate “aggression” vs. “fear” categories.
- Results
o Cannot evaluate with need for clarity in MM and analysis
- Discussion
o Limitation of active coping responses in selection of behavioural criteria in test
o Often using assumptions/statements that are not validate by any evidence (current study or literature) – i.e. general welfare, training methods, enrichment, relationship between mood/welfare and utility in EAI
o

The paper has high quality ideas/concepts. However, it is often presented in a fragmented way. Focusing on sentence structure and certain key words (e.g. mitigating vs. overriding/overshadowed; perspective of humans vs. human perspective) will greatly improve the readability.
Author Response
This is an excellent concept and likely also an excellent scientific study. The lack of clarity in the methods and certain sections of writing can be addressed to make a great publication. The authors should truly be applauded for addressing the mental domain of equine welfare in relation to type of work.
Many thanks for your comments and your very relevant review. We took into account all your advices in the pdf (please see changes tracks in the manuscript).
There is significant need for improvement of how the JBT methods are described, as such it is so unclear that it inhibits interpretation of methodology. Because of this it is hard to truly evaluate the statistical approaches, however, there is some question within their modelling if combined models/covariates were appropriately included. The authors need to address the confounding of utilizing testing restricted to evaluation of active coping methods, as well as interaction between site and quality of the human-animal interactions in addition to their focus on environmental/nutritional management being a driver of site differences. These challenges with the paper should be addressed before publication to ensure the paper conveys their hard work and very pertinent approach to an area of equine behaviour and welfare that much needs this type of support and investigation.
In addition to in-line comments in the pdf, there are a few general comments here:
- Lack of clarify on JBT testing leads to inability to truly assess scientific rigor and appropriateness of subsequent sections (analysis, results, discussion)
HHR and JBT methods have been reviewed with more details
- Time budget methods and analysis needs to be included in methods/results not as a small mention in discussion
Time budget mention has been removed
Introduction
- Sentence grammar in introduction needs attention
English grammar has been reviewed by a native English speaker
- Introduction should be re-written for clarity of concepts and relation between the concepts presented and the aims of the study. As is, it jumps around a lot and is a bit hard to follow. There is a LOT of great information within it and would love to see that come through for the readers.
Parts of the introduction have been re-written. The present study framework is about work influence on horses long lasting affective state, also define as mood in the literature. Cognitive judgement bias and reaction towards human measurements can thus be used to infer the mood of animals to better understand their welfare.
- Are they measuring compromised state related to apathy? (see intro last paragraph in predictions)
In the present study, we did not measured apathy as it requires a multifactorial approach (e.g., measures of the spontaneous behaviour and posture of horses in their home environment, evaluation of their responsiveness to their environment). But previous studies showed that EAI horses were less interactive than riding school or sport horses in human-horse relationship tests. This outcome may highlight a compromised welfare that could be related to apathetic state or lower motivation to interact with humans. Thus, we predicted that EAI-RS horses may be less interactive than RS horses during human-horse relationship tests, as a reflection of compromised welfare. The predictions have been rephrased.
Methods
- 3 sites – site 1 large EAI load
- Description of sites/work needs clarification. For example, EAI seems very similar between all sites, and similar for RS, however, how it is presented is not similar and makes the reader have to think/question where the differences are (e.g. in horse handling/tacking for RS. One consideration maybe to re-write relative to outcomes of focus being together (e.g. EAI described in general then differences by site, then RS in general and differences by site, then types of riders) rather than current approach of describing by site. Not a necessary edit, but something to reflect on when reviewing how this section appears to support a reader connecting with your methods
Description of work has been edited with EAI described in general then differences by site, then RS in general and differences by site.
- Why was no assessment of individual personality performed?
Individual personality could be influenced by work or horses could have been selected on their temperament for EAI or RS work. It would have been difficult to disentangle the influence of personality on reaction towards human and cognitive judgment.
- HHR
o Scores are biased towards active coping styles and terminology is not representative of exclusivity. Suggest renaming “negative” scores of D +E to “defensive” (which doesn’t necessarily capture the ambiguity of ear position in D) or maintain consistency with results section of “aggressive”. However, standing with ears back and not engaging is not a exclusive sign of aggressive behaviour in horses, so proceed with labelling scores with caution as to the implications and assumptions for the readers
We corrected terminology throughout the paper. Scores D + E are aggressive behaviours and F + G fear behaviours. We added the threatening head posture for the D score. Aggressive behaviours are characterized by movements towards the experimenter whereas fear behaviours are related to movements in the opposite direction of the experimenter.
- JBT – needs major revisions for clarity of methods
o Description of training phase needs clarity of bucket pattern differing between horses or within the same horse’s training, P/N bucket differentiation relative to sensory systems of the horse.
We added details in JBT description. The buckets were always the same (same black colour and size) between horses and within horses’ training, regardless of bucket location (P / N / NP / M / NN). We also added video as supplementary materials.
- Analysis
o Add section headings within analysis, and a statement of the purpose of each analysis (e.g. “To investigate XXX prediction/hypothesis, a XXX model was used) prior to details of the models
Done
o needs clarification, which relates to needs in methods section.
- What is the purpose and use of the adjusted latency scores?
To avoid biases caused by differences in baseline running speeds due to size and/or age of individuals, raw latencies recorded to reach the ambiguous positions were transformed into scores, according to Mendl et al. (2010)
o ACT
- Please provide assessment of relationship between left/right ACT
There was a positive relationship between the behaviour of horses during ACT-left & ACT-right. We decided to group scores in ACT left and right in the results to enhance clarity.
- consider a model comparing positive vs. negative valanced reactions as the lack of score differences with current approach to combining scores questions if the chosen “grouping” of scores is reflective of the underlying intention of “positive vs. negative” reactions, particularly with the non-exclusivity of behaviours in separate “aggression” vs. “fear” categories.
We considered a model comparing positive (A+B) vs negative (D+E+F+G) reactions. This model leads to different results as aggressive and fear behaviours were not displayed in the same way according to site and work. In horses, aggressive behaviours are related to validated altered welfare indicators (e.g. back problems, negative judgment bias) whereas fear behaviours do not. Fear behaviours may reflect another perception of humans than aggressive behaviours.
- Results
o Cannot evaluate with need for clarity in MM and analysis
Done
- Discussion
o Limitation of active coping responses in selection of behavioural criteria in test
we added a paragraph on coping styles
o Often using assumptions/statements that are not validate by any evidence (current study or literature) – i.e. general welfare, training methods, enrichment, relationship between mood/welfare and utility in EAI
we followed your advice and discussed only results from the study.
Round 2
Reviewer 1 Report
Comments and Suggestions for Authors
The amended version of the manuscript is substantially improved however I still have further comments
The Table legends for S2 & 3 still do not provide sufficient information to be read independently from the text of the manuscript.
Line 60 omit ‘study’
Line 172 supplementary feed
Line 173 At sites 2 and 3
30+20+16 = 66 You studied only 30 horses. It needs to be made clearer in the text that at sites 1, 2 and 3 that x of 30, y of 20 and z of 16 horses were studied. I think x=13, y=10 and z=7 from Table 1.
Table 1 Age to 1 decimal place would be more biologically plausible!
Nb of meals/day Nb is not defined
Comment on huge variability in amount of work performed – Total 1.5 and 11 hours a week; proportion of EAI work for EAI-RS horses was from 9% to 86% of their working time
How might this have influenced results? Your sample size was small. This needs to be discussed.
Line 202 ‘which varied between riding sites’ would be better as ‘and varied among riding sites’
Line 222 ‘Compare to EAI,’ This would be better as ‘In contrast to EAI, RS riders ….’
Lines 221 & 223 At (not in) Site 1 At sites 2 and 3
Line 234 ‘
• The approach contact test (ACT) [34,35], where the experimenter…. Amend to ‘In the approach contact test (ACT) [34,35], the experimenter…. ‘
Line 244 amend to ‘from the horse’s left side first and then from the horse’s right side …’
Line 249 ‘At least half a day after the ACT (the experimenter removed from the pasture and retrieved the halter), ….’ What does this mean?
Line 291 led not lead
Line 348 Days 4 to 7 …… each horse’s learning performance
Line 369 All horses were not tested on the same day, depending on each horse training performance. Change to ‘All horses were not tested on the same day, because this depended on each horse’s training performance.’
Line 382 running is not an appropriate word for a horse. Do you mean trotting?
Lines 393, 426, 437 sites (not site)
Line 612 ‘….perception of horses of their environmental..’ change to ‘perception of horses to their environmental..’
Line 646 ‘Riding could increase dorsal problem and pain..’ What does this mean?
Please review the literature concerning lameness and other musculoskeletal pain in riding school horses.
Line 663 ‘Facility significantly impacted the ….’
Line 694 e.g. short and tensed reins, reining [12] - what does this mean? And you need to make it clear that these were personal observations. Do you mean that riders were observed to shorten the reins and increase rein tension and use rein cues to direct movement from side to side?
Line 703 ‘showed more fragmented actions’ What is meant by fragmented actions?
Given the high prevalence of orthopaedic pain in riding school horses I think that you need to discuss how this might influence the results.
A major limitation of the study is an absence of assessment of the presence or absence of pain – a single osteopathic examination annually is not sufficient.
Please discuss what could be done to mitigate fear & aggression.
Comments on the Quality of English Language
I have made suggested changes
Author Response
Reviewer #1
The Table legends for S2 & 3 still do not provide sufficient information to be read independently from the text of the manuscript.
Table S2: Median latency (in seconds) in the judgment bias test to go to the bucket for the positive (P) and negative (N) learnt locations and the three ambiguous locations: near positive (NP), middle (M), and near negative (NN); (N=30).
Table S4: Relationship between human perception by horses measured as positive, indifferent, aggressive or fear reactions for each horse in the approach contact and the halter tests and judgment bias adjusted latency within the three ambiguous bucket locations: near positive (NP), middle (M), and near negative (NN); (N=30).
Line 60 omit ‘study’
L60: done
Line 172 supplementary feed
L172: done
Line 173 At sites 2 and 3
L174: done
30+20+16 = 66 You studied only 30 horses. It needs to be made clearer in the text that at sites 1, 2 and 3 that x of 30, y of 20 and z of 16 horses were studied. I think x=13, y=10 and z=7 from Table 1.
Details have been added in the text: 13 of 16 in site 1, 10 of 20 in site 2 and 7 of 30 in site 3.
Table 1 Age to 1 decimal place would be more biologically plausible!
Thanks. Change to 1 decimal has been done
Nb of meals/day Nb is not defined
Nb: number is defined in the legend.
Comment on huge variability in amount of work performed – Total 1.5 and 11 hours a week; proportion of EAI work for EAI-RS horses was from 9% to 86% of their working time. How might this have influenced results? Your sample size was small. This needs to be discussed.
We agreed. We added some details in the methods explaining the sample size and table S3 show Spearman correlations details on the relationship between workload and JBT.
L162-166: All horses involved in riding school and EAI activities (EAI-RS) across the three sites were included in the study. To create a comparable control group, we then selected horses from the same sites exclusively involved in RS activities, ensuring that their individual characteristics (e.g., age, sex) were as close as possible to those of the EAI-RS group.
It should be noted that in France there are no large facilities exclusively dedicated to EAI or with many horses involved in EAI (e.g. Grandgeorge et al, 2024, reference #14) which limits the possibility to have a large sample of EAI horses on the same site. In addition, the same horses are often chosen by the riding teachers / facility managers to be involved in EAI and beginners riding lessons, which could have an impact on these horses well-being. This has been added in the discussion:
L695-703: Variability in workload quantity between sites could be related to managers horses’ choice criteria according to the intended activity. RS managers reported personality and physical characteristics as main criteria and EAI managers put more emphasis on horses with positive relationship with humans [14]. This suggested that the same horse could be often chosen and, hence, had a high workload that may influence its welfare. However, the present study did not highlight a relationship between workload quantity or proportion of EAI activity and behaviours in the JBT suggesting the influence of the quality of working conditions.
Line 202 ‘which varied between riding sites’ would be better as ‘and varied among riding sites’
L207: done
Line 222 ‘Compare to EAI,’ This would be better as ‘In contrast to EAI, RS riders ….’
L227: In contrast to EAI
Lines 221 & 223 At (not in) Site 1 At sites 2 and 3
L226-228: done
Line 234 ‘
- The approach contact test (ACT) [34,35], where the experimenter…. Amend to ‘In the approach contact test (ACT) [34,35], the experimenter…. ‘
L239: thanks, done
Line 244 amend to ‘from the horse’s left side first and then from the horse’s right side …’
L249: done
Line 249 ‘At least half a day after the ACT (the experimenter removed from the pasture and retrieved the halter), ….’ What does this mean?
The halter test was not done successively after the ACT. It has been rephrased (L255).
Line 291 led not lead
L295: done
Line 348 Days 4 to 7 …… each horse’s learning performance
L346: done
Line 369 All horses were not tested on the same day, depending on each horse training performance. Change to ‘All horses were not tested on the same day, because this depended on each horse’s training performance.’
L368: done
Line 382 running is not an appropriate word for a horse. Do you mean trotting?
Yes! L380: done
Lines 393, 426, 437 sites (not site)
Done
Line 612 ‘….perception of horses of their environmental..’ change to ‘perception of horses to their environmental..’
L612: done
Line 646 ‘Riding could increase dorsal problem and pain..’ What does this mean? Please review the literature concerning lameness and other musculoskeletal pain in riding school horses.
We added more details on the impact of riding school on horses’ health disorder and we added references.
L648-651: Working conditions during riding school lessons, in particular the potential emotional tensions induced during these lessons, the rider's posture and actions, and/or the quality or fitting of the working equipment, could increase lameness, dorsal problems and other musculoskeletal pain [11,44], which had been related to aggressive reactions in human-horse relationship tests.
Line 663 ‘Facility significantly impacted the ….’
L667: done
Line 694 e.g. short and tensed reins, reining [12] - what does this mean? And you need to make it clear that these were personal observations. Do you mean that riders were observed to shorten the reins and increase rein tension and use rein cues to direct movement from side to side?
Riders in riding site 2 and 3 were told by the riding teachers to have short and tense reins. L701-706: Horses in riding site 1 were mostly involved in EAI sessions that lasted an hour with low amount of time spent in grooming, mounting was mostly associated with a lift and riding teachers promoted lengthen reins actions. Horses in riding sites 2 and 3 were involved in RS activities mainly, with more restrictive techniques (e.g. short and tensed reins, reining, personal observation, which has been related to high and hollow neck postures and horses’ back disorders [11]).
Line 703 ‘showed more fragmented actions’ What is meant by fragmented actions?
L715: fragmented actions i.e. grooming stops with breaking contact between the brush and the horse’s body)
A major limitation of the study is an absence of assessment of the presence or absence of pain – a single osteopathic examination annually is not sufficient.
This has been added in the limitations
L750-753: Given the high prevalence of back and orthopedic pain in riding school horses, it would be interesting to add measurement of horses’ neck posture or assess the presence of pain which is also related to cognitive alteration in horses [59].
Please discuss what could be done to mitigate fear & aggression.
L767-769: Changes in working techniques, such as the use of positive training (e.g. positive training) may mitigate the proportion of horses displaying aggressive or fear behaviours in HHR and showing pessimism in JBT.
Reviewer 2 Report
Comments and Suggestions for Authors
The revisions have been done with mindfulness to the current study and greatly improved the comprehension of the methods and results. This is an excellent topic for readers, and is now presented in an easy to follow format.
When reviewing the proofs, please pay attention to appropriate section headings within the statistics, results, and discussion (i.e. use of numerical headings is the appropriate format vs. current indented bullet points) and consistency of terminology "riding facility" vs. "riding site"
Author Response
The revisions have been done with mindfulness to the current study and greatly improved the comprehension of the methods and results. This is an excellent topic for readers, and is now presented in an easy to follow format.
Many thanks for your valuable comments.
When reviewing the proofs, please pay attention to appropriate section headings within the statistics, results, and discussion (i.e. use of numerical headings is the appropriate format vs. current indented bullet points) and consistency of terminology "riding facility" vs. "riding site"
Done